# Design and Determination of Aerodynamic Coefficients of a Tail-Sitter Aircraft by Means of CFD Numerical Simulation

Emmanuel Alejandro Islas-Narvaez [1], Jean Fulbert Ituna-Yudonago [1,*], Luis Enrique Ramos-Velasco [1], Mario Alejandro Vega-Navarrete [1] and Octavio Garcia-Salazar [2]

1   Department of Aerospace Engineering, Metropolitan Polytechnic University of Hidalgo, Boulevard Acceso a Tolcayuca 1009, Ex Hacienda San Javier, Tolcayuca 43860, Hidalgo, Mexico
2   Aerospace Engineering Research and Innovation Center, Faculty of Mechanical and Electrical Engineering, Autonomous University of Nuevo Leon, San Nicolás de los Garza 66455, Nuevo Leon, Mexico
*   Correspondence: jituna@upmh.edu.mx; Tel.: +52-464-112-02-81

**Abstract:** Vertical take-off and landing (VTOL) aircraft have become important aerial vehicles for various sectors, such as security, health, and commercial sectors. These vehicles are capable of operating in different flight modes, allowing for the covering of most flight requirements in most environments. A tail-sitter aircraft is a type of VTOL vehicle that has the ability to take off and land vertically on it elevators (its tail) or on some rigid support element that extends behind the trailing edge. Most of the tail-sitter aircraft are designed with a fixed-wing adaptation rather than having their own design. The design of the tail-sitter carried out in this work had the particularity of not being an adaptation of a quad-rotor system in a commercial swept-wing aircraft, but, rather, was made from its own geometry in a twin-rotor configuration. The design was performed using ANSYS SpaceClaim CAD software, and a numerical analysis of the performance was carried out in ANSYS Fluent CFD software. The numerical results were satisfactorily validated with empirical correlations for the calculation of the polar curve, and the performance of the proposed tail-sitter was satisfactory compared to those found in the literature. The results of velocity and pressure contours were obtained for various angles of attack. The force and moment coefficients obtained showed trends similar to those reported in the literature.

**Keywords:** flight dynamics identification; autonomous vehicles; model reduction; dynamic emulation; CFD simulation





## 1. Introduction

The development of autonomous and unmanned aerial vehicles has been a key approach adopted by the aerospace industry in recent years. These new vehicles incorporate flight capabilities that allow them to operate in different scenarios and fulfill multiple operational purposes. Much of the qualities that these aircraft possess are attributed to their designs, which involve defining the flight mission, selecting the vehicle configuration, studying the known theory of similar aircraft, and relating the physical properties of each design to the results obtained using mathematical and computational tools.

Aircraft design processes can vary depending on the type of aircraft or part to be designed. There are some studies in the literature in which the design processes are different from the previous ones. Chung et al. [1] performed the design, fabrication, and flight testing of a flying wing UAV electric thruster. The design process consisted of defining the performance requirements, stall speed, maximum speed, rate of turn, cruise altitude, absolute ceiling, and radius. The wing loading and associated power loading were derived based on defined performance requirements. The aerodynamic and stability design of the aircraft began with the given wing reference area. Then, the shape and configuration of the aircraft were determined. XFLR5 and DATCOM software were used to build the 3D model to estimate the zero-lift drag coefficient and induced drag factor of the aircraft.The

proposed flying wing UAV was made from composite materials. Asim [2] proposed the different stages to follow when designing an aircraft. The stages consist of defining the design requirement, the mission that the aircraft must perform, the load to be supported, the configuration of the main components of the aircraft, and the avionics, stability, and control systems. However, the paper does not present any particular design or analysis. Espinal-Rojas et al. [3] proposed design and construction processes for a prototype of an unmanned aerial vehicle equipped with artificial vision to search for people, which include: the design of a structure with the correct morphology of the drone, which allows for taking in-flight images; the design of a communication system that allows these images to be sent to a ground base; and, finally, the design and implementation of a control and artificial vision system for the correct flight and the identification of the existence of people on the ground. Chu et al. [4] carried out a review on the design, modeling, and control of morphing aircraft. Within this work, they proposed three main design steps: configuration design, dynamic modeling, and flight control. Rahman et al. [5] conducted a design and performance analysis of unmanned aerial vehicles delivering aid in remote areas. In this study, they proposed a design procedure consisting of: the definition of the mission requirement, the explanation of the design, the detailed design of the vehicle components, the configuration of the navigation system, and the final design. Similar to the previous authors, Khan et al. [6] proposed a design procedure that includes the following steps: the preliminary design, which includes the selection of the aerodynamic profile of the wing and empennage and the sizing and design of components; the determination of stability characteristics; and the detailed design. Likewise, Kontogiannis et al. [7] proposed four main design stages: conceptual design, preliminary design, aerodynamic analysis and optimization, and final design.

CAD modeling is the most important stage in aircraft design as it streamlines the design process and improves the visualization of sub-assemblies, parts, and the final product. It also enables an easier and more robust design documentation, including the geometry, dimensions, and bills of materials. This stage is complemented by FEA and CFD analysis, which allow for simulating engineering designs made with CAD to assess their characteristics, properties, feasibility, and profitability. Tyan et al. [8] performed a multidisciplinary design optimization of a tailless unmanned combat aerial vehicle (UCAV) using global variable fidelity aerodynamic analysis. The UAV design was developed using the CAD, Dassault CATIA. Three different geometric models (baseline, low-fidelity, GVFM) were designed and numerically simulated by CFD ANSYS Fluent to determine the properties of lift, drag, polar curve, and pressure distribution on the wing. Harasani [9] carried out the design, construction, and testing of an unmanned aerial vehicle. The analysis of the aerodynamic properties was performed using the commercial design software TORNADO. For stress analysis, ANSYS FEA was used, and an EXCEL spreadsheet was used for the performance analysis and stability derivatives. Iqbal and Sullivan [10] presented an integrated approach to a conceptual UAV design, which includes: the design stage with CAD software, structural analysis with FEA software, and aerodynamic analysis with CFD software. Sobester and Keane [11] described, illustrated, and demonstrated a conceptual UAV design system through a specific design case study. It was shown that commercial CAD tools can be integrated into the design process from the conceptual level, where, as parametric geometry engines, they can play the important role of providing the models required by the various lines of multidisciplinary analysis. This allows for the use of CFD and FEA solutions in the early design stages. Flynn [12] demonstrated techniques that can lead to optimized designs by using 3D solid modeling CAD software and then performing elementary CFD simulations and simple stress FEA analysis. In addition, he demonstrated how computer-optimized designs can be produced economically using state-of-the-art rapid prototyping and rapid production machines based on additive manufacturing techniques. Chowdhur et al. [13] carried out an aircraft design through the wing subsystems and a VTOL multi-rotor subsystem, involving the design of wing modules, fuselage, engines, stabilizers, and the battery. Then, they optimized the efficiency

parameters $L/D$, autonomy, weight, and cost, but they carried out an aerodynamic analysis for different conditions—polar curve or performance analysis—with respect to similar designs. Katon and Kuntjoro [14] designed a new/original modular UAV platform in which different combinations of modules were assembled to create a fixed-wing UAV and a quadrotor UAV. A 3D CAD modeling was carried out to conceive the basic configuration of the modules and the fixing mechanisms between modules. Three-dimensional CAD modeling was also used to illustrate the optimized module and assembly configurations. The use of the CATIA parametric CAD system in aircraft aerodynamic design was investigated and demonstrated by Ronzheimer [15]. It was shown that a parametric CAD system can act as a geometry generator, producing a clean geometry input for CFD simulations. MohamedZain et al. [16] designed and simulated a small-sized unmanned aerial vehicle (UAV) using 3DEXPERIENCE software. The design process of the frame parts involved many methods to ensure that the parts can meet the requirements. Todorov [17] performed a structural and modal analysis of a wing box structure using numerical simulation. The research was based on the finite element analysis of the aircraft wing. The first six natural frequencies and mode shapes were obtained. Benaouali and Stanisław [18] presented a new procedure for the aircraft wing structure design process. The originality of this procedure lies in the complete automation of the design process, the complexity in the considered case of the wing structure, and the applicability to different types of wings through parametric modeling. The design process involved both geometric modeling and structural analysis through the integration of two commercial CAD and CAE tools. The aerodynamic and stability characteristics of a fixed-wing MPU RX-4, a flying wing UAV, were studied by Bliamis et al. [19]. The preliminary design phase was performed and the aerodynamic performance, as well as the stability and control behavior, were evaluated using semi-empirical correlations, which were specifically modified for winged drones light flywheels, and the CFD tool.

Among the properties that must be determined in the analysis and optimization phase of an aircraft design, the most important are the aerodynamic properties (lift and drag force, lift coefficient, drag coefficient, rolling moment coefficient, pitching moment coefficient, yawing moment coefficient, angle of attack, polar curve proportionality constant, speed for maximum lift–drag ratio, stall speed, and Oswald coefficient). These properties allow for validating the flight conditions suggested in the preliminary design stage. This analysis is performed using CFD simulation. The simulation of aerodynamic elements by means of CFD, in the aeronautical industry, represents a fundamental pillar in the development of new technologies and optimization of existing models. Menter et al. [20] determined the polar curve of an aircraft by numerical simulation. The results were successfully validated with experimental data. Şumnu et al. [21] demonstrated the effects of shape optimization on the missile performance at supersonic speeds. The aerodynamic coefficients of drag and lift under different Mach numbers and different angles of attack were investigated numerically by means of CFD ANSYS Fluent software. Lao and Wong [22] carried out an aerodynamic study through CFD simulation that allowed for knowing the behavior of the aerodynamic properties (drag, lift, and moment coefficients) of an aircraft under different angles of attack. Combining the results from the lift and drag coefficients, it could be summarized that the lift–drag relationship is greatly improved by the ground effect. Salazar and Lopez [23] carried out an aerodynamic study of a wing in two and three dimensions by means of CFD numerical simulation and reported the lift and drag coefficients for various angles of attack. They did not obtain results of aerodynamic moments or perform variations in the yaw angle. Manikantissar and Geete [24] presented a conceptual design of an aircraft and performed CAD drawing and CFD simulation with ANSYS CFX to determine the distribution of pressures and lift and drag forces for multiple angles of attack and geometric taper values. They did not determine any properties related to the aerodynamic performance. Lammers [25] carried out the modeling of a commercial aircraft. CFD simulation was performed to determine the pressure, velocity, density, and temperature fields of the air around the airplane, without an engine. Lift and

drag forces were also calculated. The results were validated by means of experimental data. Gu et al. [26] performed CFD numerical modeling and simulations on a commercial aircraft to obtain the polar curve at different Mach numbers and with different numerical models. Kosík [27] performed the modeling and CFD simulation of a twin-engine aircraft to determine pressure contours, streamlines, and drag and lift coefficients versus the angle of attack by comparing ANSYS Fluent and OpenFOAM results with experimental data.

Apart from the aerospace field, CFD simulation also allows for predicting the aerodynamic behavior of land vehicles and wind turbines. In land vehicles, the most frequent cases are those of numerical studies focused on competition vehicles equipped with spoilers and other elements that allow them to improve their grip on the ground. Cravero and Marsano [28] used Ansys CFX code to investigate the ground effect on an open-wheel race car. This work focused on demonstrating the reliability of a RANS model to study the flow around the unprotected rotating wheels of a racing car, where the interaction with the multi-element inverted wing is very noticeable, including the entire front of the car, of an actual F1 model from the year 2000. Ravelli and Savini [29] modeled an F1 car and carried out CFD simulation with open source software to determine pressure contours and streamlines. Wang et al. [30] carried out a CFD simulation of an F1 car to obtain the aerodynamic forces and, later, the results were compared with wind tunnel tests data for different wind velocities. In wind turbines, the aerodynamic flow around the turbine blade is complex in nature and difficult to analyze through experiments. By using CFD, different aerodynamic properties of a blade can be easily analyzed. Fernandez-Gamiz et al. [31] used CFD simulation to analyze the aerodynamic behavior of Gurney flaps and microtabs in a wind turbine. Different CFD simulations were made to compute the lift-to-drag ratio for several angles of attack. In addition, the CFD simulations allowed for the sizing of the passive flow control devices based on the airfoil's aerodynamic performance. Aziz et al. [32] performed a numerical and experimental investigation of the drag and lift forces at low Reynolds numbers and at different angles of attack to optimize the design of a turbine blade. bCFD simulation was mainly applied to analyze the factors that are not possible to visualize in real time, the factors on which the drag coefficient is dependent, and how it should be minimized. Ung et al. [33] investigated the aerodynamic performance of a vertical axis wind turbine with an endplate design. CFD simulation was carried out using the sliding mesh method and the k-$\omega$ SST turbulence model on a two-bladed NACA0018 VAWT. The aerodynamic performance of a VAWT with offset, symmetric V, asymmetric, and triangular endplates was analyzed and compared against the baseline turbine.

Throughout the works reported above, it can be pointed out that most of the works related to the design of tail-sitter aircraft adapted a fixed-wing instead of making their own design, and that most of them only showed basic aerodynamics properties and did not take into account the contribution of aerodynamics moments or the slip angle variation. In addition, there are few publications where the design and simulation methodology is reported in a complete way. The present work develops: (a) a geometric design; (b) an aerodynamic analysis of a tail-sitter UAV through the use of CAD and CFD computational tools; (c) a performance evaluation of the final design through some efficiency parameters that give an idea of the quality of the design.

## 2. Material and Method

### 2.1. Tail-Sitter Design

The proposed tail-sitter design was based on a moderately swept trapezoidal wing. It is a twin-engine prototype capable of taking off and landing on rigid supports that are attached to the wing and extend below the trailing edge [34–37]. The different design steps are shown in Figure 1.

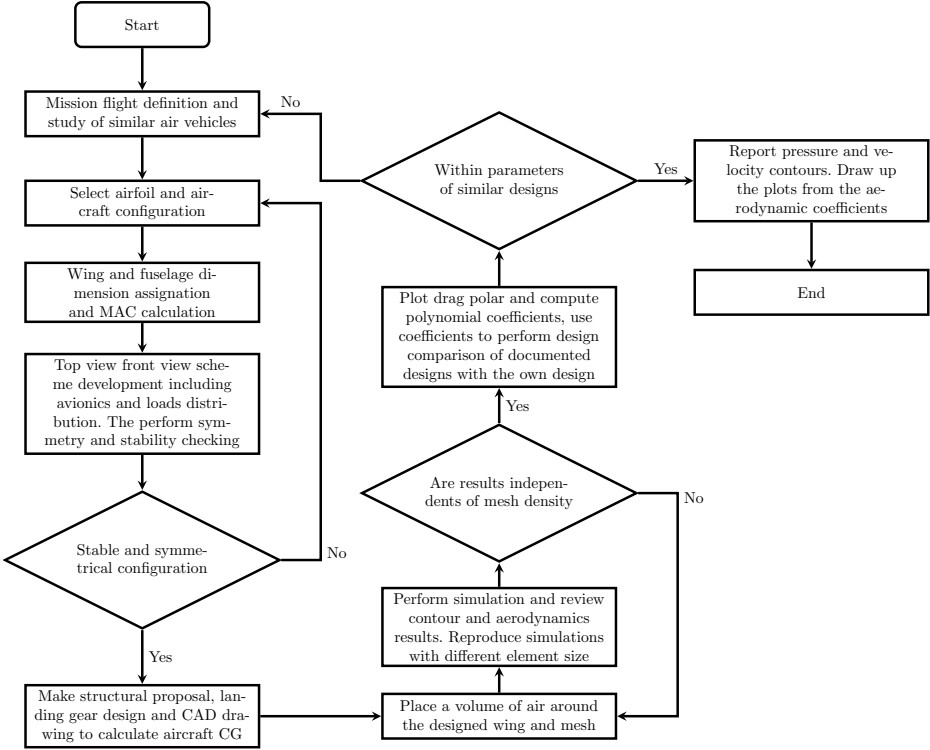

**Figure 1.** Flow chart of the wing design procedure.

### 2.1.1. Tail-Sitter Wing Sizing

This section consists of determining the sizes of the wing. In order to achieve this, an aerodynamic profile based on the MH60 profile was proposed [38], which is a profile frequently used in the design of composite wing or delta wing aircraft. For the central section of the wing that corresponds to the fuselage, re-scaling was applied to the original profile, with the purpose of having a greater internal space that allows for distributing the loads of the components installed inside the aircraft. Equations (1)–(4) were used to find the wing sizes shown in Table 1.

$$b_s = (L_l + L_{sc}) - \frac{b}{2} \tag{1}$$

$$\Lambda_r = \arctan\left(\frac{L_{ed}}{b - F_w}\right) \tag{2}$$

$$L_{ed} = C_{RW} - C_{TW} + T_{ed} \tag{3}$$

$$\Lambda_p = \arctan\left(\frac{T_{ed}}{b - F_w}\right) \tag{4}$$

In order to add some directional stability to the design, sweep angles were added to the wing. The first sweep angle was formed between the root and tip leading edges $\Lambda_{ba} = 7.34$; in the same way, the second sweep angle was formed between the trailing edges of the wing $\Lambda_{bs} = 16.00$.

The elevators are the mechanical elements responsible for the rotational movements that the tail-sitter can carry out. These elements were proposed with the dimensions described in Table 1. These mechanical elements were selected according to their dynamic effectiveness, which, in contrast to other documented designs [37], are similar in the moment coefficients that are produced. In the same way, the proportion of the elevator chord with respect to the mean wing chord was taken into account. The current proposal contemplates 20%, which is slightly below the most frequent proportion of 40%.

**Table 1.** Dimensions of tail-sitter wing.

| Wing Description (Symbol) | Value [Units] |
|---|---|
| Fuselage width ($F_w$) | 0.16 [m] |
| Half wing length ($b_s$) | 0.76 [m] |
| Taper ($\lambda$) | 0.76 |
| Root chord ($C_{Rw}$) | 0.25 [m] |
| Tip chord ($C_{Tw}$) | 0.19 [m] |
| Dihedral angle ($\Gamma$) | 3 [°] |
| **Elevator description (Symbol)** | |
| Elevator length ($b_e$) | 0.35 [m] |
| Elevator root chord ($C_{Re}$) | 0.044 [m] |
| Elevator tip chord ($C_{Te}$) | 0.044 [m] |
| Reference chord ($C_{ref}$) | 0.3908 [m] |
| **Fuselage description (Symbol)** | |
| Center section length ($L_{sc}$) | 0.35 [m] |
| Side section length ($L_l$) | 0.044 [m] |
| Offset length ($L_D$) | 0.044 [m] |
| Fuselage length ($L_{Fu}$) | 0.3908 [m] |
| **Motor support description (Symbol)** | |
| Motor base ($m_b$) | 0.0798 [m] |
| Airfoil part ($a_{part}$) | 0.162 [m] |
| Superior segment ($S_{seg}$) | 0.068 |
| Elevon angle ($\delta_{elv}$) | 71 [°] |
| Inferior segment ($i_{seg}$) | 0.046 [m] |
| Superior separation ($s_{sep}$) | 0.068 [m] |
| Inferior separation ($i_{sep}$) | 0.090 [m] |
| Thickness ($t_s$) | 0.015 [m] |

Figure 2a presents the elementary scheme of the wing with the dimensions of the root chord, the tip chord, half wing length, regressive sweep angle, and progressive sweep angle whose values are previously reported in Table 1. Figure 2b shows the detailed scheme of the wing, including the elevator dimensions. Values of these dimensions are previously reported in Table 1.

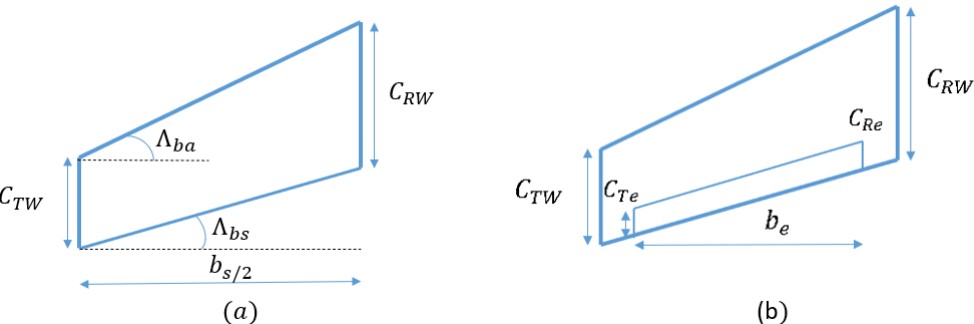

**Figure 2.** Wing schemes: (**a**) Elementary scheme, (**b**) Detailed scheme.

Once the dimensions of the wing are known, the mean aerodynamic chord can be determined. The mean chord is a parameter of great importance in the study of aerodynamic properties and in the placement and distribution of loads when carrying out weight and balance tasks. The mean aerodynamic chord can be calculated using Equation (5) [39].

$$MAC = \frac{2}{S} \int_0^{\frac{b}{2}} c(y) dy \qquad (5)$$

where $S$ is the wing area, $d(y)$ is the function that describes the chord change with respect to the wing station, and $c(y)$ is the half wing chord that varies linearly from root to tip. The half wing chord can be calculated by the following Formula (6)

$$c(y) = C_{Rw} - \frac{2(C_{Rw} - C_{Tw})y}{b} \tag{6}$$

Carrying out the analytical development and simplifications of Equation (5) and substituting each quantity for its value yields $MAC = 0.2214$ [m]. This value mainly serves as a reference point for the placement of loads on board the tail-sitter.

### 2.1.2. Fuselage Sizing

In this section, the fuselage sizing is carried out from the middle of the aircraft. The corresponding scheme is shown in Figure 3a, and the dimensions are detailed in Table 1.

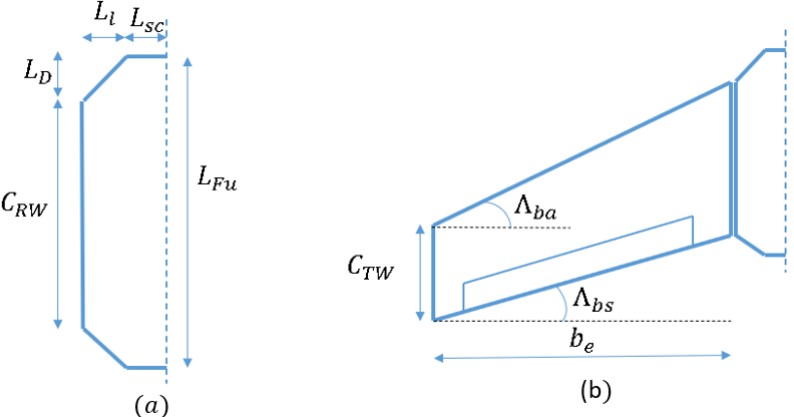

(a)

(b)

**Figure 3.** Fuselage scheme: (**a**) Half fuselage isolated, (**b**) Fusselaje and wing assembly.

The fuselage of the tail-sitter involves a longer central segment where avionics elements can be placed more freely. The design is based on an octagonal shape in which the two vertical sides have a height equivalent to the root chord of the semi-wings in order to facilitate the assembly of both elements as can be seen in Figure 3b.

### 2.1.3. Global Tail-Sitter Scheme

The global tail-sitter scheme is first presented in the front view in Figure 4 to visualize the dihedral angle ($\Gamma$) of the wings. This angle is responsible for modifying certain properties of lateral stability by adding a certain degree of robustness to the roll angle against disturbances or gusts of wind. Because greater angles can significantly displace the center of gravity, causing instability in the vertical flight, for the present prototype, three degrees of the dihedral angle were considered, as previously described in Table 1. The second global scheme of the tail-sitter is presented in top view in Figure 5 to visualize the sweep angle ($\Lambda$) of the wings. The distribution of tail-sitter components (wing, fuselage, and elevator) is symmetrical to the centerline where the center of gravity (CG) is located. The colored points represent instruments and hardware locations in the tail-sitter.

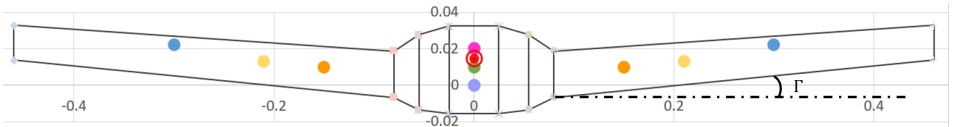

**Figure 4.** Front view of global tail-sitter scheme.

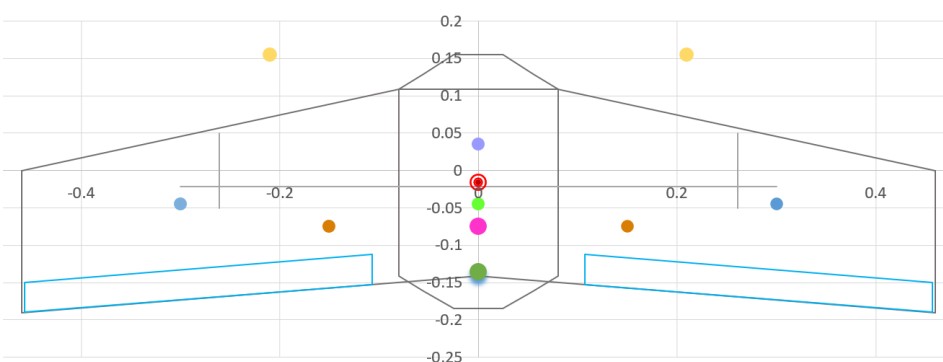

**Figure 5.** Top view of global tail-sitter scheme.

### 2.1.4. Structural Design of Tail-Sitter

The structural design of the tail-sitter components was based on the different models developed in the literature [40–42]. To carry out this structural design, firstly, two wing spars were proposed, whose positions are shown in Figure 6. The front spar is located at 13 percent of the middle chord from the leading edge, and the center spar at 58 percent. Each of these spars is made of balsa wood and has a square cross section with rounded edges. The side of the square measures 11 [mm] and the rounding at each vertex is 2.77 [mm].

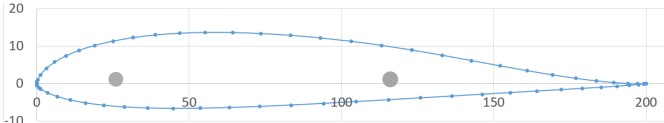

**Figure 6.** Side view of spar location.

The second step consisted of sizing the ribs and the motor support. Eight ribs made of balsa wood with a thickness of 3 [mm] were proposed in each half-wing, as well as a motor support (in each half-wing) made of square-shaped ABS polymer with a side of 27.5 [mm] and a total length of 358 [mm]. The bottom of each motor support has two diverging branches separated by an angle of 71°, between which, the elevator is mounted. The end of the support has a straight shape that is 114 [mm] long, with the same cross section as the trunk, which serves as landing gear. The motor support scheme is shown in Figure 7 and the dimensions and magnitudes of each part of the motor support are previously described in Table 1.

The spacing between ribs can be divided into two parts: the first distributes the first four ribs equidistantly from the root to the motor support, resulting in one rib every 12% of the half span. The second part joins four ribs from the motor support to the wingtip, resulting in one rib every 17% of the half span. Details of the spacing of each rib are presented in Table 2.

**Table 2.** Rib spacing.

| Position [mm] | Chord [mm] | 1st Spar [mm] | 2nd Spar [mm] | Comments |
|---|---|---|---|---|
| 0 | 250.00 | 32.50 | 145.00 | |
| 25 | 246.05 | 31.98 | 142.71 | Internal elevator |
| 72.5 | 238.55 | 31.01 | 138.36 | |
| 120 | 231.05 | 30.03 | 134.01 | Motor support |
| 185 | 220.07 | 28.70 | 128.05 | |
| 250 | 210.52 | 27.36 | 122.10 | |
| 315 | 200.02 | 26.03 | 116.15 | |
| 380 | 190.00 | 24.70 | 110.20 | External elevator |

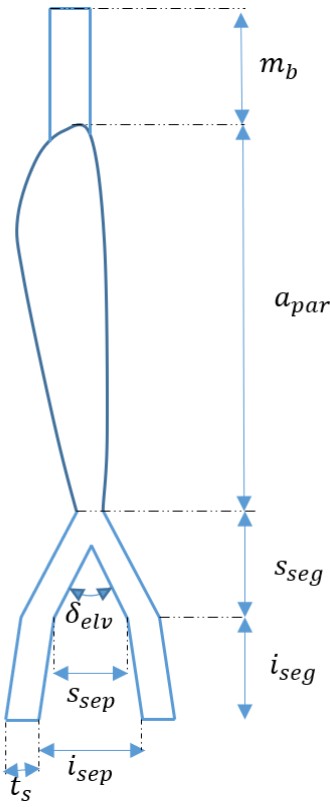

**Figure 7.** Motor support scheme.

The third step was to dimension the fuselage structure. The structure is made up of two ribs of balsa wood with a thickness of 3 [mm] and a cord of 340 [mm], both separated by 52 [mm]. These ribs are installed on two spars made of balsa wood, located in the same positions as those of the wings, with dimensions identical to the previous ones.

The fourth step involved drawing each component in CAD using ANSY Spaceclaim software. The assembly drawing of all of the structural components is shown in Figure 8.

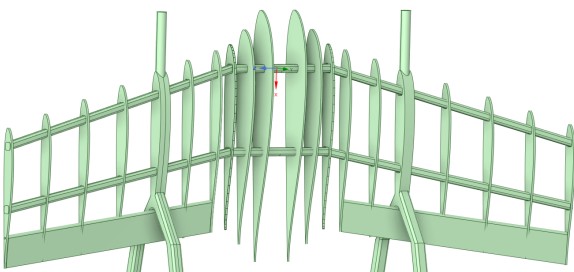

**Figure 8.** Detailed drawing of the tail-sitter structural arrangement.

The fifth step consisted of designing and locating the boxes that must house the various electronic components of the aircraft, such as Pixhawk flight computer, Raspberry pi main processing computer, battery, GPS antenna, electric motors, electronic speed controllers, servos, and ground supports. Each one of these boxes has the same dimensions as the physical component in terms of volume and assigned density, which allows it to match the mass to the real component. The dimensions and material of each component are described in Table 3. Regarding the location of each component, components such as flight computers, the battery, and the GPS antenna were placed above the longitudinal axis in a position close to the trailing edge because, in this way, a symmetry of loads can be guaranteed and the center of gravity can be positioned behind the center of pressures. The electric motors were placed at the tip of the ground support because, in this position, the propellers have

free movement and the elevators can take advantage of the propwash generated by each propeller. The speed controllers usually have short cables that connect to each electric motor; therefore, these devices were placed outside the fuselage, especially in front of each elevator symmetrically from each other. The servos were also placed outside the fuselage, ahead of each elevator at a distance where the wing's convexity allowed them to protect the servos. Their location was approximately at the center of the elevator. The supports were placed at the same distance from the center of the wing, where each electric motor was located. The set of components can be seen in Figure 9.

**Table 3.** Dimensional characteristics of the components.

| Name | Material | Length [mm] | Width [mm] | Depth [mm] | Mass [kg] |
|---|---|---|---|---|---|
| Pixhawk | Plastic and semiconductors | 50 | 15.5 | 81.5 | 0.038 |
| Rasp berry pi | Plastic and semiconductors | 88 | 58 | 19.5 | 0.046 |
| Battery | Lithium polymer | 105 | 33 | 24 | 0.08 |
| Antenna GPS | Ceramics and semiconductors | 54 | 54 | 15 | 0.03 |
| Electric motor 1 | Metal | 27.5 | 27.5 | 30 | 0.033 |
| Electric motor 2 | Metal | 27.5 | 27.5 | 30 | 0.033 |
| ESC 1 | Semiconductors | 55 | 28 | 7 | 0.01 |
| ESC 2 | Semiconductors | 55 | 28 | 7 | 0.01 |
| Servo 1 | Plastic | 40 | 20 | 38 | 0.022 |
| Servo 2 | Plastic | 40 | 20 | 38 | 0.022 |
| Ground support 1 | ABS polymer | 40 | 20 | 38 | 0.17 |
| Ground support 2 | ABS polymer | 40 | 20 | 38 | 0.17 |

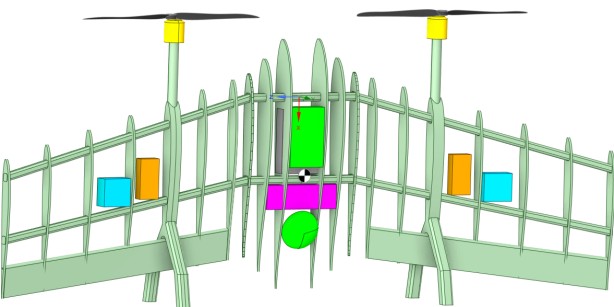

**Figure 9.** Detailed drawing of the tail-sitter structural arrangement and components.

The sixth step was to choose the material for the skin of the aircraft and its coating. A shrink-wrap film known as "monokote" was chosen, which is 1 [mm] thick and has a weigh per unit area of 0.061 [kg/m$^2$]. This material covers the entire external surface of wings and fuselage as can be seen in different views in Figure 10.

The last step consisted of calculating the center of gravity of the prototype designed. The calculation was based on the formulas proposed by Sandonis [43].

$$x_{CG} = \frac{\sum_i m_i x_i}{M} \tag{7}$$

$$y_{CG} = \frac{\sum_i m_i y_i}{M} \tag{8}$$

$$z_{CG} = \frac{\sum_i m_i z_i}{M} \tag{9}$$

where $m_i$ is the individual mass of each component, $M$ is the total mass of the components, $(x_i, y_i, z_i)$ is the center of gravity of each component whose reference point is the origin of the aircraft's nose coordinates, and $(x_{CG}, y_{CG}, z_{CG})$ is the position of the center of gravity. The mass of each component was obtained from the manufacturers' data sheet. For the motor support, the mass was determined from the CAD Space Claim. The value of the mass of each component is previously described in Table 3. Thus, the total mass of the tail-sitter was 0.942 [kg], which corresponds to the sum of the masses of all of the components. The $(x_i, y_i, z_i)$ of each component was determined according to the location of each one.

These three parameters allowed us to calculate the coordinates of the center of gravity $(x_{CG}, y_{CG}, z_{CG})$, whose values in [m] were, respectively, [0.178, $7 \times 10^{-7}$, and 0.0152].

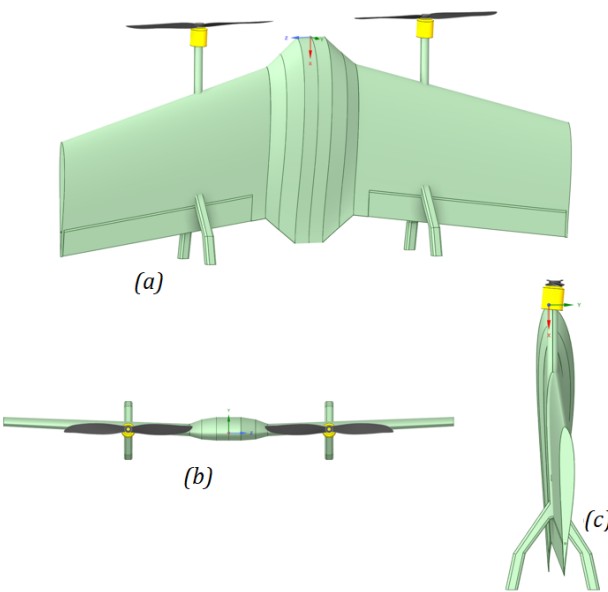

*(a)*

*(b)*

*(c)*

**Figure 10.** Virtual prototype of the tail-sitter: (**a**) Top view. (**b**) Front view. (**c**) Side view.

*2.2. Determination of Aerodynamic Coefficients by Means of CFD*

2.2.1. Mathematical Model for CFD Simulation of Tail-Sitter

The mathematical model used to develop the numerical solution was carried out by means of a 3D model of the equations of mass, momentum, and turbulence in steady state.

The Spalart Allmaras model was chosen to analyze the turbulent behavior of the airflow around the tail-sitter. This model is commonly used in aerodynamic problems and is very suitable for predicting pressure gradients within the boundary layer [22]. The $y^+$ corresponding to this model is very low ($y^+ < 5$), which allows it to accurately predict aerodynamic properties close to the boundary layer compared to 2-equation models such as $k - \epsilon$, whose $y^+$ is higher ($y^+ < 30$) [44–46].

The following considerations were taken into account for the development of the mathematical model:

- The simulation takes place in an isothermal environment.
- Air is the working fluid and is assumed to be viscous, incompressible, and behave like an ideal gas.
- The gravity force acts in a direction normal to the intrados of the wing.
- The density and viscosity of the fluid remain constant.

The governing equations were formulated compactly as follows:
Mass conservation equation:

$$\nabla(\rho v) = 0 \tag{10}$$

Momentum conservation equation:

$$\nabla \cdot (\rho v T) = \nabla\left(\frac{\lambda}{C_p}\nabla T\right) + \Psi \tag{11}$$

Turbulent viscosity equation for Spalart Allmaras model formulated by [47]:

$$u_j\frac{\partial \tilde{v}}{\partial x_j} = C_{b1}[1 - f_{t2}]\tilde{S}\tilde{v} + \frac{1}{\sigma}\Big\{\nabla \cdot [(v + \hat{v})\nabla\tilde{v}] + C_{b2}|\nabla\tilde{v}|^2\Big\}$$

$$-\left[C_{w1}f_w - \frac{C_{b1}}{k^2}f_{t2}\right]\left(\frac{\tilde{v}u}{d}\right)^2 + f_{t1}\Delta U^2 \tag{12}$$

where $\nu = \frac{\mu}{\rho}$ is the molecular kinematic viscosity and *rho* is the density, $\nabla \cdot [(\nu + \tilde{\nu})\nabla\tilde{\nu}]$ is the classical diffusion operator, $\sigma$ is the Prandtl turbulence number, $C_{b2}|\nabla\tilde{\nu}|^2$ is a non-conservative term, and $C_{b1}[1 - f_{t2}]\tilde{S}\tilde{\nu}$ is the turbulence destruction term in which

$$\tilde{S} \equiv S + \frac{\nu}{k^2d^2}f_{v2} \tag{13}$$

where $d$ is the distance between the nearest surface and $S$ is defined as

$$S = \sqrt{2\Omega_{ij}\Omega_{ij}} \tag{14}$$

where $\Omega_{ij}$ is the rotation tensor given by

$$\Omega_{ij} = \frac{1}{2}\left(\frac{\partial u_i}{\partial x_j} - \frac{\partial u_j}{\partial x_i}\right) \tag{15}$$

$$f_{v2} = 1 - \frac{\chi}{1 + \chi f_{v1}} \tag{16}$$

$$f_{t1} = C_{t1}g_t exp\left(-C_{t2}\frac{\omega_t^2}{\Delta U^2}\left[d^2 + g_t^2d_t^2\right]\right) \tag{17}$$

The turbulence destruction term is contemplated by

$$\left[C_{w1}f_w - \frac{C_{b1}}{k^2}f_{t2}\right]\left(\frac{\tilde{\nu}}{d}\right)^2 \tag{18}$$

where

$$f_w = g\left[\frac{1 + C_{w3}^6}{g^6 + +C_{w3}^6}\right] \tag{19}$$

$$g = r + C_{w2}(r^6 - r) \tag{20}$$

$$r \equiv \frac{\nu}{\tilde{S}k^2d^2} \tag{21}$$

$$f_{t2} = C_{t3}exp(C_{t4}\chi^2) \tag{22}$$

The constants included in the turbulence model are described in Table 4.

**Table 4.** Constants included in the turbulence model.

| Constant | Value |
|---|---|
| $\sigma$ | 2/3 |
| $C_{b1}$ | 0.1355 |
| $C_{b2}$ | 0.622 |
| $k$ | 0.41 |
| $C_{w1}$ | $C_{b1}/k^2 + (1 + C_{b2})/\sigma$ |
| $C_{w2}$ | 0.3 |
| $C_{w3}$ | 2 |
| $C_{v1}$ | 7.1 |
| $C_{t1}$ | 1 |
| $C_{t2}$ | 2 |
| $C_{t3}$ | 1.1 |
| $C_{t4}$ | 2 |

### 2.2.2. Geometric Model

The geometric model for the numerical simulation as shown in Figure 11a is composed of two domains: the tail-sitter domain and the air enclosure domain. This model was carried out using the CAD ANSYS DesignModeler, in which, the domain of the tail-

sitter was exported from the prototype built in CAD ANSYS SpaceClaim and the air enclosure domain was drawn based on [22,48]; the dimensions are described in Table 5. The computational domain shown in Figure 11b surrounded the tail-sitter aircraft, which has the same dimensions as the design stage. However, motors and motor support were not considered since the fluid dynamic analysis focuses on the aerodynamic forces.

**Table 5.** Air enclosure dimensions.

| Description | Value [Units] |
| --- | --- |
| Length | 830 [mm] |
| Width | 300 [mm] |
| Height | 60 [mm] |

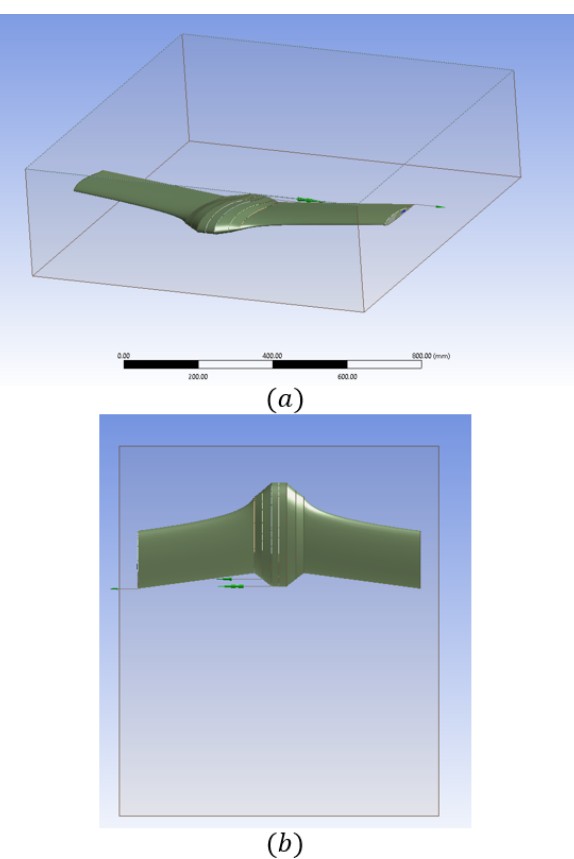

(*a*)

(*b*)

**Figure 11.** Computational domains for CFD simulation: (**a**) Isometric view, (**b**) Top view.

### 2.2.3. Boundary Conditions

The walls of the air enclosure domain, as shown in Figure 11, were the boundaries for the numerical simulation. The front wall was considered as the air flow inlet boundary, wose condition was a velocity inlet with a magnitude of 16 [m/s] (value used as a condition to design the tail-sitter). The rear wall was considered as the outlet boundary, with a gauge pressure outlet condition of 0 [Pa] (since the absolute pressure is equal to the atmospheric pressure). The other walls (upper, lower, right, and left sides) were considered as symmetry boundary conditions in a similar way to that of [22,48]. This allowed the air velocity to be maintained above zero in the area close to the walls to avoid the hydrodynamic boundary layer effect.

Unlike the wall boundary condition, the symmetry boundary condition allows for simulations with reduced air domains. The use of this symmetry condition allowed us to eliminate the boundary layer effect due to the reduction in the air velocity close to the wall [22,49]. Boundary conditions are resumed in Table 6.

**Table 6.** Boundary conditions.

| Location | Boundary Type | Value [Units] |
|---|---|---|
| Front face | Velocity inlet | 16 [m/s] |
| Rear face | Gauge pressure outlet | 0 [Pa] |
| Upper face | Symmetry | - |
| Lower face | Symmetry | - |
| Right face | Symmetry | - |
| Left face | Symmetry | - |

2.2.4. Mesh Independence Analysis

The mesh independence study was carried out with the purpose of finding the mesh size limit at the point where the predictions obtained from the simulation are independent at greater mesh size reductions [50]. This analysis was preceded by the meshing process of the air enclosure domain. This choice is justified by the fact that the goal pursued by the CFD simulation is to determine the aerodynamic properties throughout the external surface of the tail-sitter. However, it is not necessary to consider the tail-sitter domain in the mesh. To perform this process, an unstructured mesh was made in order to reduce the number of nodes and, consequently, the computational time of the analysis. The meshing was carried out using the ANSYS Meshing software included in the ANSYS Workbench platform [51]. The meshed computational domain is shown in Figure 12. It is possible to see a global view of the mesh domain, as well as a region close to the aircraft where polygons conform a fine mesh in the zone that makes contact between the air domain and aircraft skin. The mesh refinement was made using body sizing with capture curvature and capture proximity in order to reduce the size of the mesh close to the aircraft skin. The distance of the first border node was 0.10949 [mm]. This value was checked with free software [52] using properties such as the density $\rho$, viscosity $\mu$, mean aerodynamic chord $MAC$, y plus $y^+$, and air velocity $U_\infty$. The distance calculated by means of this software was 0.116074268 [mm], which is close to that of the generated mesh. In the zoom box, it is possible to see the size ratio between the grid near the wing and the grid at the bottom boundary.

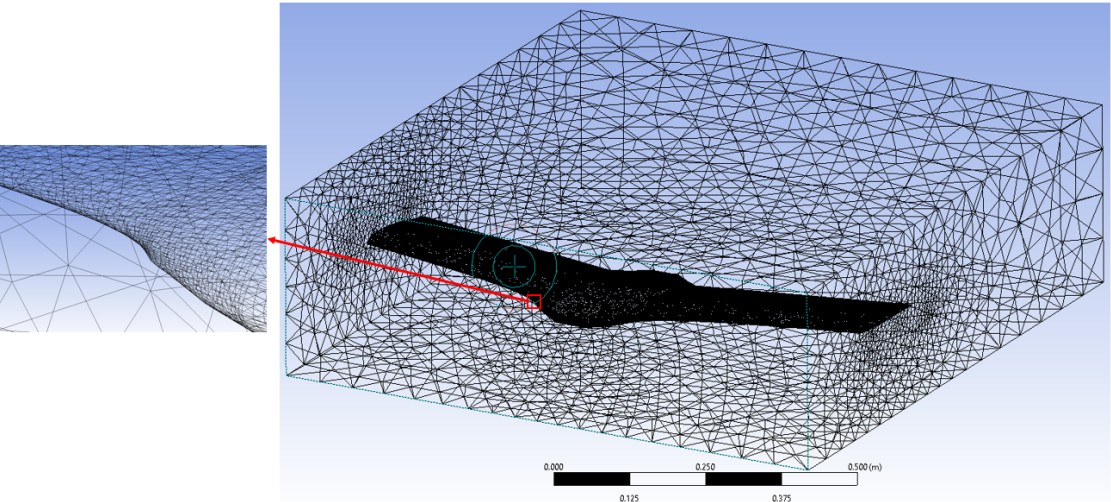

**Figure 12.** Mesh domain.

The mesh independence analysis consisted of determining the behavior of the yaw moment coefficient ($C_l$) with respect to the number of elements of the mesh. The angle of attack of zero degrees was selected to place the computational domain in a cruise flight position as can be seen in Figure 12. To perform this analysis, four different mesh sizes were selected. The characteristics of each mesh are described in Table 7.

**Table 7.** Statistics in the study of mesh independence.

| Characteristics | Mesh Sizes | | | |
| --- | --- | --- | --- | --- |
| | **M1** | **M2** | **M3** | **M4** |
| Nodes | 59,909 | 66,701 | 74,868 | 84,543 |
| Elements | 328,718 | 350,992 | 403,549 | 455,174 |
| Mesh metric Skewness | | | | |
| Minimum | $2.138 \times 10^{-4}$ | $2.293 \times 10^{-4}$ | $2.5457 \times 10^{-4}$ | $3.340 \times 10^{-4}$ |
| Maximum | 0.7415 | 0.7979 | 0.9251 | 0.9795 |
| Average | 0.2322 | 0.2328 | 0.2338 | 0.2331 |
| Standard deviation | 0.1219 | 0.1225 | 0.1237 | 0.12156 |

Each mesh was simulated in a steady state using CFD ANSYS Fluent from the Workbench platform. The flow and turbulence models, as well as the boundary conditions described in the previous section, were used. Regarding the pressure–velocity coupling algorithm, the coupled segregation algorithm was used, since it is the most appropriate algorithm to predict the behavior of pressure and velocity on the wing surface of aircraft [25]. For the discretization scheme of momentum and turbulent viscosity equations, second-order upwind was used for more precise results and, furthermore, upwind is the most appropriate scheme for discretizing the turbulent flow model [25]. Regarding the residuals, $10^{-4}$ was used for continuity and velocity in the x, y, and z direction and $10^{-6}$ for the turbulent kinetic viscosity (nut).

The results of this analysis are presented in Figure 13. It can be seen that the yaw moment coefficient is independent of the mesh size when the element number exceeds 403,549. Therefore, it was decided to use the mesh of 455,174 elements in the present study in order to obtain more precise results.

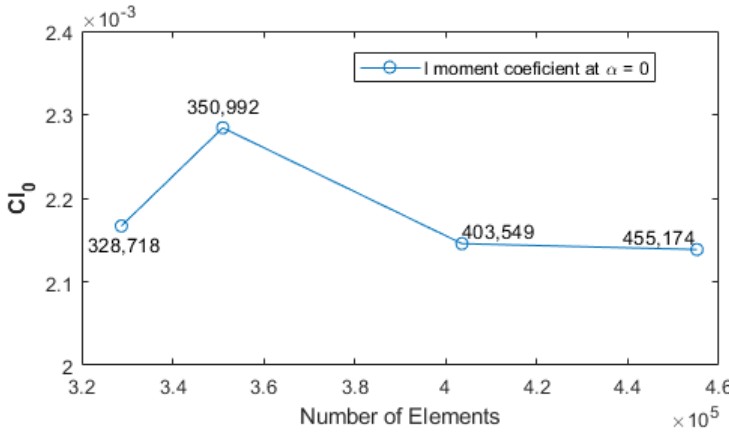

**Figure 13.** Mesh independence of moment coefficient $C_l$.

## 3. Results and Discussions

### 3.1. Validation of Tail-Sitter CFD Simulation Results

The polar curve is an algebraic representation that describes the value of the drag coefficient as a quadratic function of lift and is an invaluable resource when evaluating the efficiency of an aircraft under particular conditions [53]. This curve was used to validate the results of the numerical simulation carried out in this work. Figure 14 compares the polar tail-sitter curve produced by the results of the CFD simulation with that produced by an empirical correlation [54].

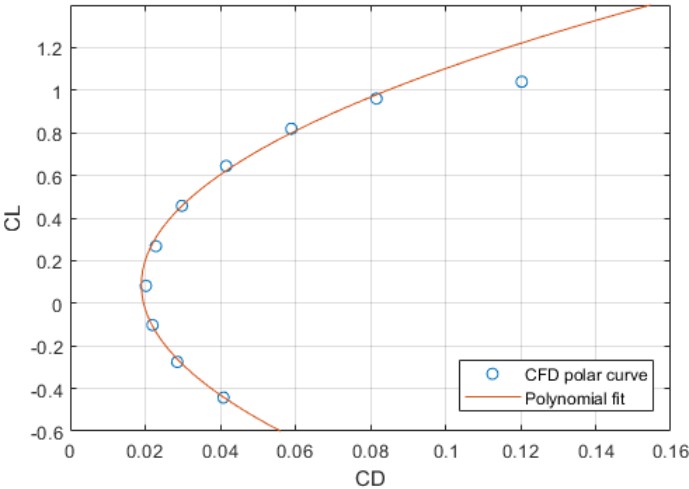

**Figure 14.** Tail-sitter polar curve: CFD vs. polynomial fit.

A similarity between the polar curve generated by the empirical correlation and that generated by numerical simulation was observed. Figure 15a,b describe the error percentages between the CFD simulation results and the polynomial fit of the lift and drag coefficients, respectively. A good approximation was observed between the two results. The maximum error for the $C_L$ is around 10%, whereas, for the drag coefficient, the error reaches 25% when the $C_D$ produced by the CFD simulation is equal to 0.12, but the average error is 10%, which is within a suitable range.

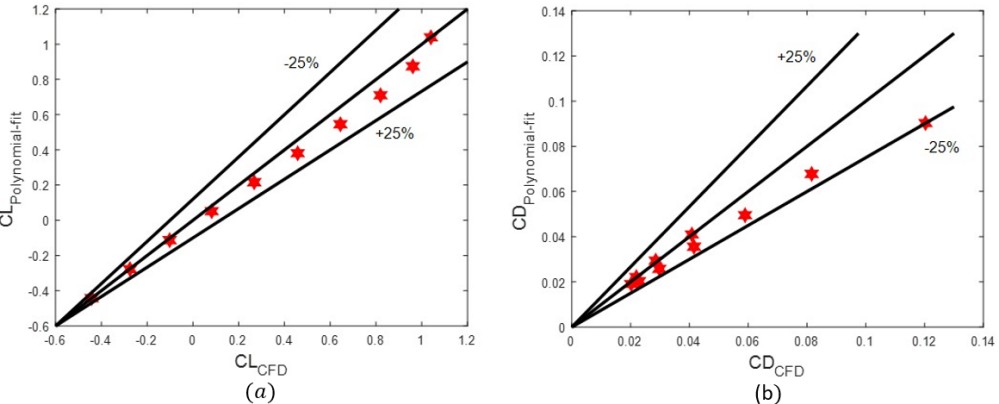

**Figure 15.** Percentage of errors between CFD and polynomial fit: (**a**) $C_L$ and (**b**) $C_D$.

Therefore, the numerical model developed in the present study can predict the aerodynamic behavior of the designed tail-sitter.

### 3.2. Aerodynamic Properties of the Tail-Sitter

The main aerodynamic properties in this study are the coefficients of forces and moments. However, to better understand the effects of aerodynamic forces on the tail-sitter at different angles of attack ($\alpha$), the pressure contours on the tail-sitter surface, as well as the airspeed distribution around the airfoil, were reviewed. The pressure contours on the extrados zones and intrados zones are shown in Figures 16 and 17, respectively. Positive pressures were mainly obtained on the extrados and negative pressures on the intrados when $\alpha = -4$. This behavior is normal because, when the aircraft is tilted down, the airflow is intercepted by the upper surface (extrados), causing greater pressure. The pressure decreases progressively when the angle of attack approaches zero, and when the angle of attack begins to increase as shown by the pressure contours when $\alpha = 4$ and $\alpha = 8$, the pressure decreases on the extrados and increases on the intrados, allowing the aircraft to

lift. The wing reaches a minimum pressure of $-133.5$ [Pa] at the beginning of the lower surface and a maximum pressure of 142 [Pa] at the leading edge, which is clearly acceptable for the zone where each pressure is located.

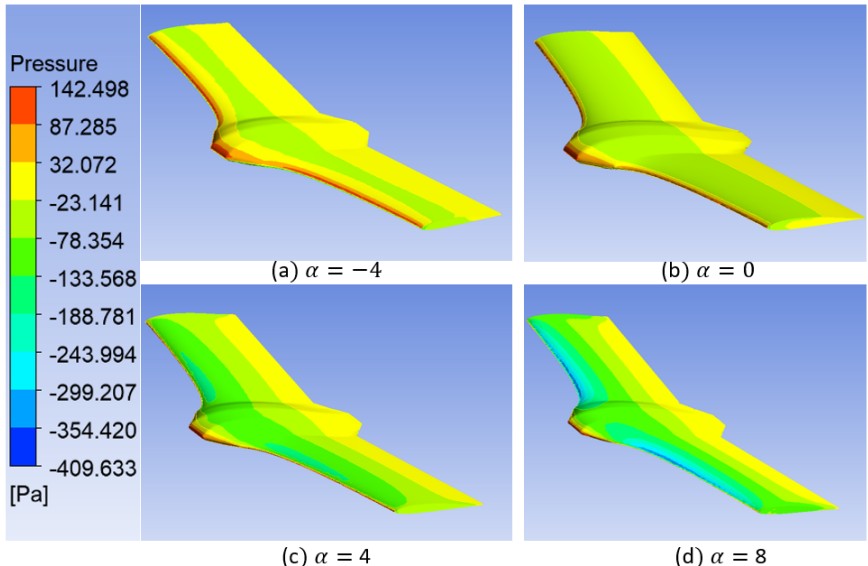

**Figure 16.** Pressure distribution on the tail-sitter surface at different angles of attack at upper surface.

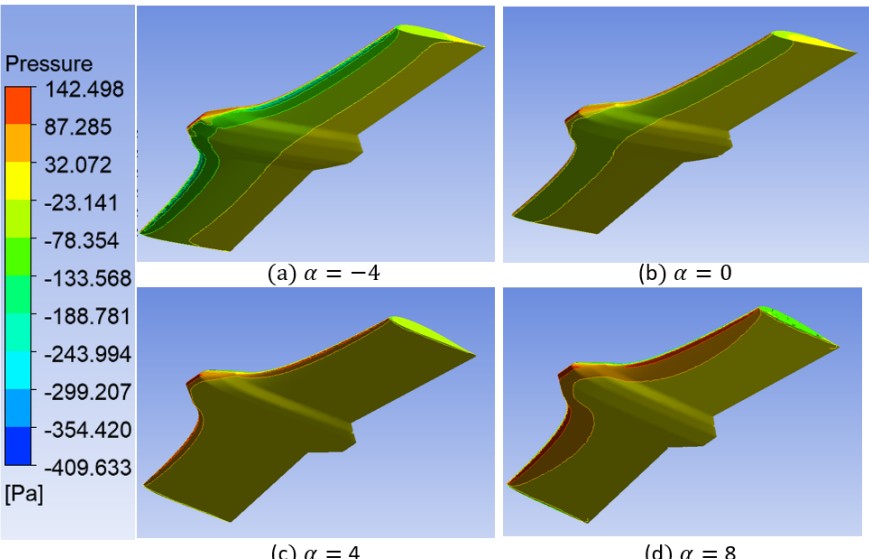

**Figure 17.** Pressure distribution on the tail-sitter surface at different angles of attack at lower surface.

The inverse of the pressure behaviors around the tail-sitter area is observed in Figure 18a–c for the velocity distributions. When the aircraft is tilted downwards, the airflow is intercepted by the upper surface (extrados), causing a decrease in the air velocity and, consequently, an increase on the lower surface. The air velocity remains almost in balance at approximately 18 [m/s] on both sides when the angle of attack approaches zero, and when the angle of attack starts to increase (see the velocity contours when $\alpha = 4$ and $\alpha = 8$), the air velocity decreases on the intrados by up to 10 [m/s] and increases on the extrados by up to 23.3 [m/s], allowing for a better lift of the aircraft. In addition, the air velocity decreases progressively along the mid-span from root to tip. This avoids a greater wake effect at the tip.

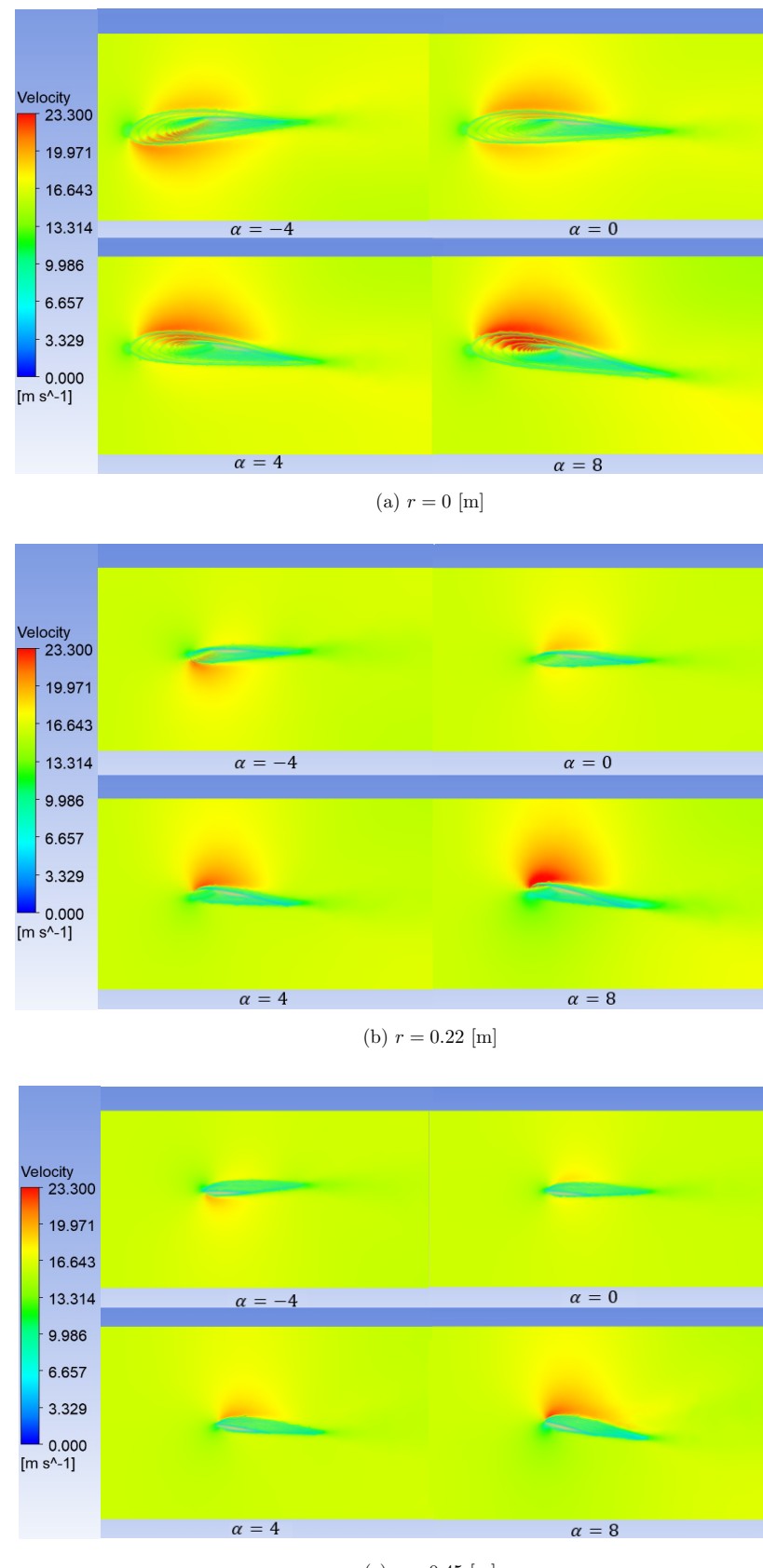

Figure 18. Velocity contours at different angles of attack: (**a**) local radius $r = 0$ [m], (**b**) local radius $r = 0.22$ [m] and (**c**) local radius $r = 0.45$ [m] .

In general, the pressure and velocity contours show reasonable behavior from an aerodynamic point of view. This once again confirms the good performance of the tail-sitter design proposed in this study.

The effects of the changes in the angle of attack and angle of slip ($\beta$) on the force coefficients and moment coefficients are analyzed below.

Figure 19a shows the behavior of the lift coefficient against the variations in both angles. The lift coefficient increases linearly from $-0.4$ to 1 as the angle of attack increases from $-6°$ to $10°$. The trend is slightly changed when the angle of attack exceeds $10°$. The variations in the angle $\beta$ do not provide significant effects on the lift coefficient since the proposed tail-sitter has a swept trapezoidal wing. This behavior matches those reported in the literature for the trapezoidal wings [1,19,55,56].

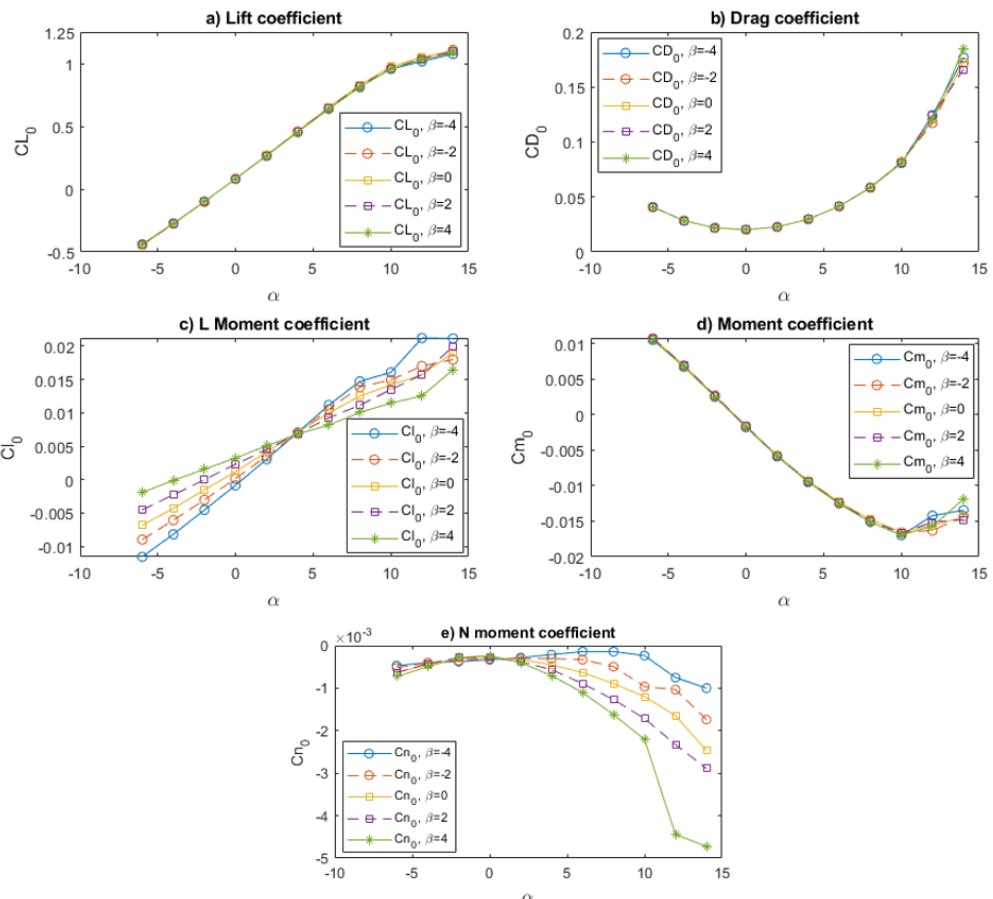

**Figure 19.** Effects of changes in angle of attack and angle of slip on the force coefficients and moment coefficients: (**a**) lift coefficient, (**b**) drag coefficient, (**c**) L moment coefficient, (**d**) M moment coefficient, (**e**) N moment coefficient.

Figure 19b shows the behavior of the drag coefficient against changes in the angles $\alpha$ and $\beta$. The drag coefficient does not increase linearly like the lift coefficient. It has a tendency of a curve of second degree, as is usual in the behavior of the drag coefficient of trapezoidal wings when the angle of attack varies; see Refs. [57,58]. Similar to the lift coefficient, variations in the angle $\beta$ do not provide a significant change in the drag coefficient. The drag coefficient reaches its minimum value of 0.02017 when the angle of attack is $0°$ and reaches 0.17 when $\alpha = 14$. This magnitude is within the adequate range for trapezoidal wings as reported in the literature [1,19,55,56].

Figure 19c describes the effects of changes in the angles $\alpha$ and $\beta$ on the moment coefficient around the $X$ axis (responsible for the roll movement). A linear increase in the L moment coefficient is observed as the angle of attack $\alpha$ increases from $-0.4°$ to

14°. In addition, increasing the angle $\beta$ modifies the slope of each line. This behavior is adequate since the modification of the slip angle causes the displacement of the center of pressure of the aircraft in a lateral direction, producing a torque that contributes to the roll movement [59].

The effects of changes in the angles $\alpha$ and $\beta$ on the moment coefficient around the $Y$ axis (responsible for the pitch movement) are described in Figure 19d. The moment coefficient $Cm_0$ decreases from 0.012 to $-0.017$ when the angle $\alpha$ increases from $-4°$ to $10°$. This behavior is due to changes in the magnitude of the center of pressure and its position in the longitudinal axis, which results in an increase in the longitudinal axis torque that contributes to the pitch motion.

Figure 19e shows the behavior of the moment coefficient around the $Z$ axis (responsible for the yaw movement) against the changes in the angles $\alpha$ and $\beta$. A decrease in $Cn_0$ was observed when the angle $\alpha$ exceeds $2°$. Furthermore, the increase in the angle $\beta$ causes an increase in $Cn_0$ when the angle $\alpha$ exceeds $2°$. This is because the modification of the angle $\beta$ causes the air to hit the leading edge of one half wing more directly than the other. For this reason, the center of pressure moves laterally, producing the torque caused by the aerodynamic moment.

For aircraft with a symmetrical wing, the moments $Cl_0$ and $Cn_0$ are usually zero when $\beta = 0$ as explained by [60] in the topics of flight mechanics. This behavior is adequately demonstrated in Figure 19c,e.

### 3.3. Performance Evaluation of the Final Design

The application of CFD numerical simulation techniques to new aircraft designs allows for the knowledge of not only their aerodynamic characteristics, but also their performance. In Table 8, the performance parameter values of the integrated and trapezoidal wing design of four aircraft, documented in the literature, are compared with those of the design carried out in the present study. The results show that the performance parameter values of the design performed in the present study are within the very acceptable range. The metrics that allow for a performance comparison with other designs are presented in Table 9.

**Table 8.** Comparison of documented designs with our own design.

| Parameter | [19] | [55] | [56] | [1] | Own Design |
|:---:|:---:|:---:|:---:|:---:|:---:|
| $W/S$ | 63.765 [N/m$^2$] | - | 21.88 [N/m$^2$] | 23.544 [N/m$^2$] | 44.004 [N/m$^2$] |
| $C_{Lmax}$ | 1.235 | 0.97 | 1.09 | 1.26 | 1.12 |
| $V_s$ | 10.3435 [m/s] | - | 6.4363 [m/s] | 6.209 [m/s] | 9.004 [m/s] |
| $V_{E*}$ | 16.864 [m/s] | - | 9.359 [m/s] | 10.7863 [m/s] | 13.099 [m/s] |
| $C_{D0}$ | 0.023 | 0.007 | 0.0195 | 0.009 | 0.0207 |
| $K$ | 0.1108 | 0.0893 | 0.0734 | 0.0516 | 0.0739 |
| $e$ | 0.546 | 0.6393 | 0.86 | 0.82 | 0.92 |

**Table 9.** Dimensional characteristics of the components.

| Metrics (Symbol) | Formula | Description |
|:---:|:---:|:---:|
| $W/S$ | $\left(\dfrac{W}{S}\right) = \frac{1}{2}\rho_\infty V_s^2 C_{LMax}$ | Wing loading required at the stall speed [1] |
| $V_s$ | $V_s = \sqrt{\dfrac{2}{\rho_\infty C_{LMax}}\left(\dfrac{W}{S}\right)}$ | Stall speed [1] |
| $V_{E*}$ | $V_{E*} = \sqrt{\dfrac{2}{\rho_\infty}\left(\dfrac{W}{S}\right)}\sqrt{\dfrac{K}{C_{D0}}}$ | Speed for the maximum lift-to-drag ratio [1] |
| $K$ | $K = \dfrac{1}{\pi ARe}$ | Induced drag factor |

## 4. Conclusions

The present work was based on performing the geometric design, determining the aerodynamic coefficients, and analyzing the performance of a tail-sitter UAV using CAD and CFD computational tools. The design methodology mainly consisted of proposing and adjusting the dimensions and properties of the wing according to wing designs reported in various research works.

To carry out the structural design, an arrangement consisting of the distribution of ribs, spars, and loads from the instrumentation of the vehicle was proposed. The structural components and the instrumentation component boxes were drawn in 3D using ANSYS SpaceClaim CAD software to obtain the virtual prototype of the proposed tail-sitter.

For the CFD numerical simulation, a computational domain of air was drawn that encloses the entire part of the geometric model of the tail-sitter. The domain was meshed in an unstructured way to minimize the number of nodes and elements. A mesh independence analysis was performed based on the yaw moment coefficient. The results show that the yaw moment coefficient is independent of the mesh size when the number of elements exceeds 403,549. Therefore, it was decided to use the mesh of 455,174 elements to obtain more precise results. Based on this mesh, numerical simulations were performed in ANSYS Fluent CFD software to determine the aerodynamic coefficients. The numerical results were satisfactorily validated with empirical correlations for the calculation of the polar curve and comparison of the performance of the proposed tail-sitter with those found in the literature.

Satisfactory results of velocity and pressure contours were obtained for various angles of attack. The results of force and moment coefficients showed trends similar to those reported in the literature. Based on these results, it was concluded that the methodology proposed in this present work is feasible in the design and determination of the aerodynamic coefficients for the tail-sitter.

The results of the performance evaluation of the final design showed that the values of the performance parameters of the proposed tail-sitter design are within the very acceptable range.

As future work, it is proposed to carry out a vertical flight stability analysis of tail-sitter trough CFD simulation in order to identify the vertical flight dynamics for the controller design. In addition, the data of aerodynamic coefficients obtained in this work will be used to carry out the dynamic model of the tail-sitter, with the purpose of designing laws of control that allow for controlling the horizontal and vertical flight phases of the vehicle.

**Author Contributions:** Conceptualization, L.E.R.-V. and M.A.V.-N.; Methodology, E.A.I.-N., J.F.I.-Y. and L.E.R.-V.; Software, E.A.I.-N., J.F.I.-Y.; Validation, E.A.I.-N., J.F.I.-Y. and L.E.R.-V.; Formal Analysis, E.A.I.-N., J.F.I.-Y. and L.E.R.-V.; Investigation, E.A.I.-N. and O.G.-S.; Writing—Original Draft Preparation, E.A.I.-N., J.F.I.-Y. and L.E.R.-V. ; Writing—Review & Editing, E.A.I.-N., O.G.-S.; Supervision, L.E.R.-V. and M.A.V.-N.; Project Administration, L.E.R.-V. All authors have read and agreed to the published version of the manuscript..

**Funding:** This research received no external funding.

**Acknowledgments:** Emmanuel A. Islas-Narvaez thanks Conacyt for the support provided during his master's studies at the Universidad Politécnica Metropolitana de Hidalgo. He also thanks Octavio Garcia-Salazar of the Aerospace Engineering Research and Innovation Center, Faculty of Mechanical and Electrical Engineering, UANL for the support received during the research stay.

**Conflicts of Interest:** The authors declare no conflict of interest.

## Nomenclature

| | |
|---|---|
| $AR$ | Aspect ratio |
| $C_{D0}$ | Zero-lift drag coefficient |
| $C_{Lmax}$ | Maximum lift coefficient |
| $e$ | Oswald coefficient |
| $K$ | Induced drag factor |
| $\rho_\infty$ | Air density |
| $S$ | Wing surface |
| $V_{E*}$ | Speed for the maximum lift-to-drag ratio |
| $V_s$ | Stall speed |
| $W$ | Aircraft weight |

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
