# Peer review of "Design and Determination of Aerodynamic Coefficients of a Tail-Sitter Aircraft by Means of CFD Numerical Simulation"

_machines, doi:10.3390/machines11010017_

Round 1
Reviewer 1 Report
This study proposes a numerical method to design and study tail-sitter aircraft in different operating conditions as various angles of attack. The research is attractive for several engineering applications. The paper is quite of good in the complex, but some requests and suggestions have been provided to increase the quality of the work.
The introduction is quite complete, however is focused mainly on the design method applied only to aero-vehicle. I suggest introducing some aerodynamical and flow dynamic analysis concerning these wings through numerical or experimental work also in other applications (as the wings installed in the racing car). Moreover, is very important to add more references on the numerical method with its possible problems. So, for example, you can add:
· -Cravero, C.; Marsano, D. “Computational Investigation of the Aerodynamics of a Wheel Installed on a Race Car with a Multi‐Element Front Wing”. Fluids, 2022, Vol. 7, Issue 6, p. 182.
· -Fernandez-Gamiz, U.; Gomez-Mármol, M.; Chacón-Rebollo, T. Computational Modeling of Gurney Flaps and Microtabs by POD Method. Energies 2018, 11, 2091.
The first paper considers the effect of the rotation of a wheel that interacts with a wing of a F1 car. The second paper adopt numerical models to analyse the effect of an appendix in a front wing for automotive application to vary the force components; this device increases the downforce of the vehicle.
The design method with the geometric representation is very extensive, but it could result difficult to be read, so I suggest to better resume the different contribution parts in a separated sub-section with a complete geometry and superimposed all the main geometrical parameters. The governing equations have been reported both for the design, both for the fluid dynamic analysis.
The CFD model description is quite complete by starting with the calculation domain presentation. Concerning the models activated, more explanation must be provided on the choice of the SA model rather thana 2 equation model. A mesh sensitivity analysis with different grid seizes have been performed but more details must be provided on the grid selected also with the local mesh size and more figure on the surface mesh. The boundary conditions should be resumed also in a table.
A validation of the CFD model has been performed showing a good agreement, but you should better report also in more details the maximum error committed with respect to the reference. The discussion of the results is complete, but several contours and diagrams are very difficult to read, I suggest enlarging and to better explain the different cases in a further preliminary section
The conclusions summarize well the main results. You should report the nomenclature sect.
Author Response
Dear Reviewer #1.
We would like to thank you for the various observations, comments or suggestions that you have formulated about our manuscript, and the will that you have to see our manuscript be improved before its publication. Below are the responses to your comments. We assure you that most of them have been reproduced in the manuscript in red characters.
Please accept our most distinguished considerations.
The authors,
Comment #1:
The introduction is quite complete, however is focused mainly on the design method applied only to aero-vehicle. I suggest introducing some aerodynamical and flow dynamic analysis concerning these wings through numerical or experimental work also in other applications (as the wings installed in the racing car). Moreover, it is very important to add more references in the numerical method with its possible problems. So, for example, you can add:
- Cravero, C.; Marsano, D. “Computational Investigation of the Aerodynamics of a Wheel Installed on a Race Car with a Multi‐Element Front Wing”. Fluids, 2022, Vol. 7, Issue 6, p. 182.
- Fernandez-Gamiz, U.; Gomez-Mármol, M.; Chacón-Rebollo, T. Computational Modeling of Gurney Flaps and Microtabs by POD Method. Energies 2018, 11, 2091.
The first paper considers the effect of the rotation of a wheel that interacts with a wing of a F1 car. The second paper adopts numerical models to analyze the effect of an appendix in a front wing for automotive application to vary the force components; this device increases the downforce of the vehicle.
Thanks for your comments and for the suggested bibliography
Both references and four others have been added in the introduction, page 4, from lines 148 to 175.
Comment #2:
The design method with the geometric representation is very extensive, but it could result difficult to be read, so I suggest to better resume the different contribution parts in a separated sub-section with a complete geometry and superimposed all the main geometrical parameters. The governing equations have been reported both for the design, both for the fluid dynamic analysis.
Thanks for your observations and suggestions. Tables related with aircraft geometrical properties have been combined in the first table (Table 1). In a similar way, figures that schematize tail-sitter dimensions have been combined in Figure 2 for wing details and in Figure 3 for fuselage details.
Comment #3:
The CFD model description is quite complete by starting with the calculation domain presentation. Concerning the models activated, more explanation must be provided on the choice of the SA
model rather than 2 equation model. A mesh sensitivity analysis with different grid sizes has been performed but more details must be provided on the grid selected also with the local mesh size and more figure on the surface mesh. The boundary conditions should be resumed also in a table.
Thanks for your suggestion
- Concerning the Spalart Allmaras model: it was chosen to analyze the turbulent behavior of the airflow around the tail-sitter. This model is commonly used in aerodynamic problems and is very suitable for predicting pressure gradients within the boundary layer [22]. The y+ corresponding to this model is very low (y+ < 5) which allows it to accurately predict aerodynamic properties close to the boundary layer compared to 2-equation models such as k − ε whose y+ is higher (y+ < 30) [44], [45] and [46].
This justification has been added in section 2.2.1, page 12, lines 300 to 305, where some details about selected viscous model were added as well as bibliographic reference that support our comments.
- Concerning the local mesh size: In Figure 12, it is possible to see a global view of the mesh domain as well as a region close to the aircraft where polygons conforms a fine mesh in the zone that makes contact between air domain and aircraft skin. The mesh refinement was made using body sizing with capture curvature and capture proximity in order to reduce the size of mesh close to aircraft skin. The distance of the first border node was 0.10949 [mm]. This value was checked with free software [52 ] using properties such as density ρ, viscosity μ, mean aerodynamic chord MAC, y plus y+ and air velocity U∞. The distance calculated by means of this software was 0.116074268 [mm] which is close to that of the generated mesh. In the zoom box, it is possible to see the size ratio between the grid near the wing and the grid at the bottom boundary.
This justification has been added in section 2.2.4, page 15, lines 355 to 360.
- Concerning the boundary conditions: The Boundary conditions and their properties have been described in Table 6.
Comment #4:
A validation of the CFD model has been performed showing a good agreement, but you should better report also in more details the maximum error committed with respect to the reference. The discussion of the results is complete, but several contours and diagrams are very difficult to read, I suggest enlarging and to better explain the different cases in a further preliminary section
Thanks for your suggestion:
- Concerning the maximum error committed with respect to the reference: In Figure 15, we made the graph of error percentages between CFD simulations and polynomial fit of the lift or drag coefficients. A good approximation is observed between the two results. The maximum error for the CL is around 10 % while for the drag coefficient the error reaches 25% when the CD produced by the CFD simulation is equal to 0.12, but the average error is 10% which is within a suitable range.
- This justification has been added in section 3.1, page 17, lines 389 to 394
- Concerning the contours and diagram: We have improved the quality of contours by increasing the size. Also, the size of the color scale has been increased.
Comment #5:
The conclusions summarize well the main results. You should report the nomenclature sect.
- Thanks for your comment: Nomenclature section have been added below Acknowledgment section

Reviewer 2 Report
The authors designed the tail-sitter aircraft based on the standard theory and assessed the aerodynamic characteristics using CFD analyses. The results will be helpful for the design process of the tail-sitter aircraft.
I will recommend your article after revisions.
The following are the comments to improve your article.
P. 7
Fig 6 and 7 show the schematic views of the vehicle.
În these figures, some colored points are shown. What is the meaning of these points?
P. 15, Sec 2.2.4
The computational domain is too small compared to the vehicle dimension.
The outer domain should be ten times larger than the span of the vehicle.
In Fig.15, the mesh dependency was investigated. The number of grid pts is essential. But the minimum length of the mesh element except the boundary layer is more critical. Then, you should add the information.
P. 17 Sec 3.2
The tail-sitter aircraft will be launched vertically.
Then, the thrust should be larger than the total weight of the vehicle.
Additionally, the stability issue will be slightly different from the horizontal flight.
I recommend you add the above issue from the viewpoint of aerodynamics.
typo
P. 6, the the > the
P. 7, Fuselaje > Fuselage
Author Response
December 05, 2022
Dear Reviewer #3.
We would like to thank you for the various observations, comments or suggestions that you have formulated about our manuscript, and the will that you have to see our manuscript be improved before its publication. Below are the responses to your comments. We assure you that most of them have been reproduced in the manuscript in green characters.
Please accept our most distinguished considerations.
The authors,
Comment #1:
Fig 6 and 7 show the schematic views of the vehicle. În these figures, some colored points are shown. What is the meaning of these points?
- Thanks for your suggestion and comments: In Figures 4 and 5 (former fig 6 and 7 ), the colored points represent instruments and hardware locations in the tail-sitter as listed in Table 3.
- This justification has been added in section 2.1.3, page 8, line 235.
Comment #2:
The computational domain is too small compared to the vehicle dimension.
- The computational domain shown on Figure 11b surrounds the tail-sitter aircraft which has the same dimensions as the design stage. However motors and motors support are not considered since the fluid dynamic analysis is focused on the aerodynamic forces.
The outer domain should be ten times larger than the span of the vehicle.
- We really appreciate your observation. But, the outer domain should be ten times larger than the span of the vehicle when the Wall boundary condition is used. However, we used symmetry boundary conditions. Unlike the wall boundary condition, the symmetry boundary condition allows simulations with reduced air domains. The use of this symmetry condition allows to eliminate the boundary layer effect due to the reduction of the air velocity close to the wall [22] [49]. It can be seen in figure 18 that there is no reduction in air velocity at the walls of the air domain, due to the use of the symmetry boundary condition.
- An explanation was added in section 2.2.3, page 14, from 337 to 340 lines.
Comment #3:
In Fig.15, the mesh dependency was investigated. The number of grid points is essential. But the minimum length of the mesh element except the boundary layer is more critical. Then, you should add the information.
- Thank you for your comments: The distance of the first border node was 0.10949 [mm]. This value was checked with free software [52 ] using properties such as density ρ, viscosity μ, mean aerodynamic chord MAC, and more y+ and air velocity U∞. The distance calculated by means of this software was 0.116074268 [mm] which is close to that of the generated mesh.
- Mesh information has been added in manuscript, especially in section 2.2.4, page 15, from 353 to 361 lines.
Comment #4:
The tail-sitter aircraft will be launched vertically. Then, the thrust should be larger than the total weight of the vehicle. Additionally, the stability issue will be slightly different from the horizontal flight. I recommend you add the above issue from the viewpoint of aerodynamics.
- We really appreciate your comment. But, the work is focused on the design and determination of aerodynamic force coefficients using CFD Numerical Simulation in cruise flight conditions. So, the vertical and horizontal flight stability will be addressed in a future work as detailed in conclusions.

Reviewer 3 Report
This work is based on performing the geometric design, determining the aerodynamic coefficients, and analyzing the performance of a tail-sitter UAV using CAD and CFD computational tools. Compared with past works which only showed basic aerodynamics properties, this work takes the contribution of aerodynamics moments or the slip angle variation into consideration. The work first makes structural design. Then using ANSYS SpaceClaim CAD to obtain a tail-sitter prototype. Numerical simulation was conducted for the calculation of the polar curve and comparison of the performance of the proposed method. The performance evaluation is designed to show the values of parameters. The article is structurally complete, and the experiments are adequate. This method could be utilized for carrying out a dynamic model in the future which is meaningful for controlling vehicles.
There are suggestions for this paper:
1. The introduction section can be streamlined by selecting the most important and representative work for presentation.
2. Section 2.1.1 has too many tables, which can be combined. Some parameters can directly be represented in the text. Besides, please pay attention to the case consistency of the symbol corner markers.
3. The process in Figure 1 is too complicated. Authors can simplify the process and mark the important parts (of the article) in the figure.
4. In Section 3.3 (Performance evaluation of the final design), it is better to introduce the metrics first through a table or add a column in Table 11 to make comments.
Author Response
December 05, 2022
Dear Reviewer #2.
We would like to thank you for the various observations, comments or suggestions that you have formulated about our manuscript, and the will that you have to see our manuscript be improved before its publication. Below are the responses to your comments. We assure you that most of them have been reproduced in the manuscript in blue characters.
Please accept our most distinguished considerations.
The authors,
Comment #1:
The introduction section can be streamlined by selecting the most important and representative work for presentation.
- We really appreciate your observation: According to first reviewer recommendations, we added a paragraph which introduced CFD applications to determine aerodynamics properties in land vehicles and wind turbines. So, it wasn’t possible to streamline the introduction content as you proposed.
Comment #2:
Section 2.1.1 has too many tables, which can be combined. Some parameters can directly be represented in the text. Besides, please pay attention to the case consistency of the symbol corner markers.
- Thanks for your suggestion: All tables related with aircraft geometrical properties (such as wing, elevators, fuselage and motor support) were combined in the first table (Table 1).
Comment #3:
The process in Figure 1 is too complicated. Authors can simplify the process and mark the important parts (of the article) in the figure.
- Thank you for your suggestion: Design steps shown in Figure 1, page 5 have been reduced for better reading.
Comment #4:
In Section 3.3 (Performance evaluation of the final design), it is better to introduce the metrics first through a table or add a column in Table 11 to make comments.
- We appreciate your suggestions: Performance metrics were introduced through a new table (Table 8) before Table 9 (Comparison of documented designs with our design). All variables are described in the nomenclature section.

Round 2
Reviewer 1 Report
All my questions and requestes have been answered and added in the revised work. Now the paper has increased its quality and it is ready for the publication in this form.
Reviewer 2 Report
The authors addressed the issues that I raised.